# Persistence of Anti-SE36 Antibodies Induced by the Malaria Vaccine Candidate BK-SE36/CpG in 5–10-Year-Old Burkinabe Children Naturally Exposed to Malaria

**DOI:** 10.3390/vaccines12020166

**Published:** 2024-02-06

**Authors:** Issa Nebie, Nirianne Marie Q. Palacpac, Edith Christiane Bougouma, Amidou Diarra, Alphonse Ouédraogo, Flavia D’Alessio, Sophie Houard, Alfred B. Tiono, Simon Cousens, Toshihiro Horii, Sodiomon B. Sirima

**Affiliations:** 1Groupe de Recherche Action en Santé (GRAS), Ouagadougou 10248, Burkina Faso; i.ouedraogo@gras.bf (I.N.); e.bougouma@gras.bf (E.C.B.); a.diarra@gras.bf (A.D.); a.ouedraogo@gras.bf (A.O.); a.tiono@gras.bf (A.B.T.); 2Department of Malaria Vaccine Development, Research Institute for Microbial Diseases, Osaka University, Suita 565-0871, Osaka, Japan; nirian@biken.osaka-u.ac.jp; 3European Vaccine Initiative, UniversitätsKlinikum Heidelberg, Voßstraße 2, 69115 Heidelberg, Germany; flavia.dalessio@euvaccine.eu (F.D.); sophie.houard@euvaccine.eu (S.H.); 4Department of Infectious Disease Epidemiology, London School of Hygiene and Tropical Medicine, London WC1E 7HT, UK; simon.cousens@lshtm.ac.uk

**Keywords:** BK-SE36/CpG, serine repeat antigen, malaria vaccine, long-term immunogenicity, *Plasmodium falciparum*

## Abstract

Information on the dynamics and decline/persistence of antibody titres is important in vaccine development. A recent vaccine trial in malaria-exposed, healthy African adults and children living in a malaria hyperendemic and seasonal area (Ouagadougou, Burkina Faso) was the first study in which BK-SE36/CpG was administered to different age groups. In 5- to 10-year-old children, the risk of malaria infection was markedly lower in the BK-SE36/CpG arm compared to the control arm. We report here data on antibody titres measured in this age-group after the high malaria transmission season of 2021 (three years after the first vaccine dose was administered). At Year 3, 83% of children had detectable anti-SE36 total IgG antibodies. Geometric mean antibody titres and the proportion of children with detectable anti-SE36 antibodies were markedly higher in the BK-SE36/CpG arm than the control (rabies) arm. The information obtained in this study will guide investigators on future vaccine/booster schedules for this promising blood-stage malaria vaccine candidate.

## 1. Introduction

Malaria remains a public health problem especially in Sub-Saharan Africa where most clinical cases (2022: 249 million, accounting for about 94% of global cases) and deaths (2022: 608,000, 95%) occur [1]. The burden has remained relatively unchanged since 2015 despite several control initiatives such as long-lasting insecticide-treated nets (LLINs), insect larvae control and indoor insecticide residue spraying (IRS); intermittent preventive therapy (IPT) and periodic chemoprophylaxis (including seasonal malaria chemoprevention, SMC); and improved diagnostics. The difficulties in malaria elimination have highlighted the plasticity of the mosquito and *Plasmodium* parasite, as well as the challenges in the implementation of these control strategies in the field [2,3,4]. In combination with various control measures, a cost-effective vaccine could hold the key to ultimately progress towards malaria eradication [3,4,5,6]. RTS,S/AS01, based on *Plasmodium falciparum* sporozoite circumsporozoite protein (CSP), was the first malaria vaccine (and the first anti-parasite vaccine) prequalified by the WHO for routine use, paving the way for GAVI and the Vaccine Alliance via UNICEF to secure a supply of 18 million doses for 2023–2025 to be used for children in regions with moderate to high malaria transmission [3,6,7,8]. Another CSP-based vaccine, R21/Matrix-M, has subsequently also been recommended by the WHO [7,9]. With reported high efficacy in phase 2 [10,11] and phase 3 trials in four African countries (Mali, Tanzania, Burkina Faso, and Kenya) [12], R21/Matrix-M obtained regulatory clearance in Ghana, Nigeria, and Burkina Faso ahead of the WHO recommendation [13]. These developments are expected to alleviate some of the limitations in the supply and distribution of a malaria vaccine [7,14]. Both vaccines aim to prevent infection by inducing immune responses that target the pre-erythrocytic (liver) stage. A promising malaria erythrocytic (blood-stage) vaccine is not yet in the pipeline. 

The release of merozoites from the liver leads to blood-stage infection, and the subsequent continuous cycles of erythrocyte invasion and rupture is responsible for clinical disease. A number of parasite proteins unique to this stage are expressed [4,15]. Challenges in developing vaccines that target the erythrocytic stage include (i) extensive genetic polymorphism/large allelic variation; (ii) the functionally redundant pathways/proteins involved in parasite invasion; and (iii) the stimulation of host immune effector mechanisms allowing the parasite to escape/subvert the host immune system [2,4,6,15,16]. Children under 5 years of age are more susceptible to severe malaria than older children; and pre-existing immunity in older children and adults impacts vaccine efficacy [17,18]. Several merozoite surface protein vaccine candidates—apical membrane antigen 1 (AMA1), merozoite surface proteins (MSP1, MSP3), and glutamate-rich protein (GLURP)—have low protective capacity despite being “sufficiently” immunogenic [19,20,21,22,23]. Notably, in recent phase 1/2a trials, reticulocyte-binding protein, RH5 [21,24], demonstrated in vitro [25] and in vivo effects on parasite multiplication rate and a 1- to 2-day delay in time to diagnosis of clinical malaria using blood-stage controlled human malaria infection (CHMI) [26]. 

Another blood-stage candidate antigen under development is *P. falciparum* serine repeat antigen-5 (SERA5)—an abundant, essential antigen that plays a role in parasite egress and molecular camouflage of merozoites [27,28,29]. Several sero-epidemiological studies in malaria-endemic areas showed that high levels of anti-SE47′ (N-terminal domain with serine repeats) or anti-SE36 (N-terminal domain without serine repeats) antibodies are associated with protection against febrile malaria, high parasitaemia, and low birthweights in semi-immune adults or children (reviewed in [30]). Previous phase 1 clinical trials evaluated the safety and immunogenicity profile of SE36 adjuvanted with aluminium hydroxide gel (BK-SE36). In a malaria-endemic area in Uganda, high antibody titres induced by the BK-SE36 vaccination were associated with a reduced risk of malaria symptoms compared with the control group [31]. Moreover, in subjects with a ≥1.92-fold increase in anti-SE36 antibody titres after vaccination, there were fewer reinfections. In those who did experience natural infection, the majority had a more than 3.35-fold increase in antibody titres after infection, suggesting immunological memory and/or a natural boosting effect [31]. In recent years, additional lessons pointed to adjuvant selection and refinements in the vaccination schedule as important factors which could contribute to increasing vaccine efficacy [32,33]. A new formulation of the BK-SE36 vaccine candidate with a Toll-like receptor 9 stimulating adjuvant, CpG-ODN (K3) showed promising safety and immunogenicity results in malaria-naïve Japanese adults [34] and more recently in malaria-exposed adults and children in Burkina Faso. K3 (ATCG ACTC TCGA GCGT TCTC) is a novel K-type synthetic oligodeoxynucleotide with unmethylated CpG motif [35,36,37]. When used concomitantly with aluminium hydroxide gel, K3 was chosen as the best Toll-like receptor 9 agonist with the highest ability to induce antibody and cellular immune responses compared to BK-SE36 alone [32]. 

In a phase 1b clinical trial, the safety and immunogenicity of the BK-SE36/CpG vaccine candidate was evaluated in three cohorts: Cohort 1: adults, 21 to 45 years; Cohort 2: children, 5 to 10 years; and Cohort 3: children, 12 to 24 months [38]. The clinical trial covered a period of 365 days after the first vaccination (Dose 1). The present study aimed to analyse the persistence of anti-SE36 antibody titres in 5- to 10-year-old children three years after receiving Dose 1. The results of this study will be important in designing future proof-of-concept trials and considering the role of a booster dose for this promising blood-stage vaccine candidate.

## 2. Materials and Methods

### 2.1. Ethical Approval

The study was conducted according to the principles of the Declaration of Helsinki (2013) and the ICH guidelines for GCP (CPMP/ICH/135/95) July 1996 (and its Revision 2, dated 9 November 2016), and in full conformity with relevant country regulations. The ethical review was performed in Burkina Faso by the ethical committee of the Ministry of Health for Biomedical Research (Ref: 2022-01-008).

### 2.2. Study Design, Site, and Participants

This study was designed to include all children aged 5 to 10 years who participated in the primary double-blind, randomised, controlled, age de-escalating, phase Ib trial and had received either BK-SE36/CpG or the rabies vaccine (control arm). The main trial and the investigational product have been described in detail [38]. Briefly, a vaccination of a full dose (1 mL) of BK-SE36/CpG contained 100 µg of recombinant SE36 antigen, 1 mg aluminium, and 1 mg CpG-ODN (K3). The rabies vaccine Verorab^®^ (Sanofi Pasteur, Lyon, France) was administered at a 0.5 mL/dose, according to the manufacturer’s recommendation.

In the primary trial, the first dose (Dose 1) was given starting in December 2018 with the last dose (Dose 3) given in May 2019, and the last visit completed in January 2020. Here, a single visit was conducted to assess the persistence/longevity of anti-SE36 antibody titres in BK-SE36/CpG vaccinees three years after Dose 1 (~32 months after Dose 3; ~2 years after the end of the primary study) (Figure 1).

Following community sensitization meetings, families whose children participated in the primary trial in Cohort 2 (5–10 years of age) were contacted and given information about the study (including objectives and procedures). Parents/caregivers of eligible children were given an opportunity to ask questions and provided written informed consent before enrolment. Written assent from children (7–10 years old) was also obtained. The inclusion criteria were: (i) child participated in Cohort 2 of the primary BK-SE36/CpG phase 1b trial; (ii) parent(s)/guardian(s) provided their written consent; and (iii) child provided voluntarily his/her assent. Exclusion criteria included receipt of blood products in the last 6 months, and the presence of any condition considered by the investigator likely to interfere with the evaluation of the immune response or any health concern that could put the participant at risk. On the same day, a questionnaire was used to collect information on malaria control measures practiced by the participant; and blood samples were obtained by venepuncture for malaria blood smear and ELISA for anti-SE36 IgG antibody titres. Blood smears were stained using Giemsa, and light microscopy was performed at Groupe de Recherche Action en Santé Parasitology Lab (Ouagadougou, Burkina Faso). 

### 2.3. Anti-SE36 IgG Antibody Assessment

ELISA measurements, outsourced to a GLP-certified testing facility (CMIC Pharma Science Co., Ltd., Hyogo, Japan), were performed using standardized methodology and expressed in arbitrary units calculated using a parallel line assay [31]. 

### 2.4. Statistical Methods 

As in the primary trial, the antibody titre fold-changes were calculated based on the differences in IgG titres measured at baseline (D0, prior to Dose 1). 

In each arm, the number and proportion of individuals with detectable SE36 IgG were reported. Geometric mean titres (GMT) were calculated. For all computations, anti-SE36 IgG antibody titres below the limit of detection were assigned a value of 8. The threshold value was set based on the mean titres calculated from a total of 60 datapoints from 20 individual malaria-naïve Japanese sera, assayed in three independent batches.

Levels of anti-SE36 IgG antibodies were compared to values obtained in the primary trial prior to Dose 1 (D0), one month after Dose 3 (D140), and one year after Dose 1 (D365). Seropositivity refers to the presence of anti-SE36 antibody titres at baseline (those with detectable antibody titres prior to Dose 1); seroconversion is defined as the proportion of vaccinated individuals whose anti-SE36 IgG titres increased >2-fold from baseline.

For comparison of titres up to Year 3 (Y3), R software version 3.6.1 was used for statistical analyses. The normality of variables was checked using the Kolmogorov Smirnov test. Numbers and frequencies were reported for categorical variables; and comparisons were undertaken using Fisher’s Exact test. Geometric means were used for quantitative data; and comparison was undertaken using the Wilcoxon rank sum test. Associations with pre-existing antibody titres and sex were examined using Prism 6.07 (GraphPad Software Inc., La Jolla, CA, USA). 

## 3. Results

### 3.1. Study Site and Population

The study site and population characteristics from Day 0 to Day 365 have been previously described [38]. In all cohorts, BK-SE36/CpG was immunogenic. Although vaccine efficacy was not a primary endpoint in the trial, an additional exploratory objective was to examine malaria incidence from Day 56 (28 days post Dose 2) until the end of the trial (Day 365). Differences in clinical malaria between treatment arms were observed in Cohort 2, in which vaccinations were completed prior to the rainy season and trial participants were not included in the seasonal malaria chemoprevention implementation in the study area, as per national guidelines [39,40]. Therefore, this cohort was selected for further follow-up, as described here. In Cohort 2, during the primary trial, the proportion of subjects with detectable IgG titres in the vaccine group after Dose 2 was 100% (n = 30, geometric mean (GM) fold change in titre: 58.4 (95% CI, 40.5–84.2)) vs. 67% (n = 10/15, GM fold change in titre: 0.9 (95% CI 0.6–1.5)) in the control group. Differences in clinical malaria (defined as ≥5000 parasites/µL of whole blood + fever, ≥38 °C) between treatment arms are shown in Figure 2 [38]. 

Twenty-four months following the end of the primary trial (=32 months after Dose 3 or 3 years after Dose 1), 44 out of 45 children were enrolled for a cohort study to evaluate the persistence of anti-SE36 antibodies in vaccinees (BK-SE36/CpG vaccine arm, N = 29 and control arm, N = 15; one child from the BK-SE36/CpG vaccine arm could not be located). This study was conducted in March 2022, during the dry season (low malaria transmission season) in Ouagadougou, Burkina Faso. No child was found to be infected with malaria parasites by light microscopy. The mean ages were 10.6 (SD = 1.1) and 11.3 (SD = 1.3) years in the vaccine and the control arm, respectively. The study participants’ characteristics are presented in Table 1.

The majority of the study participants (>50%) were reported to sleep under treated bed nets. About 20% of participants did not practice any malaria control measures. 

### 3.2. Persistence of Anti-SE36 Antibody Titres

Three years after Dose 1, 24 out of 29 (83%) children had detectable anti-SE36 total IgG antibodies in the BK-SE36/CpG group compared with seven out of 15 (47%) in the control group (Table 2) (*p* = 0.039). In the vaccine arm, a decrease from Year 1 (D365) to Y3 in the percentage of subjects with measurable titres was observed. At Y3, geometric mean antibody titres in the vaccine arm were higher than those in the control arm. Figure 3A shows the distribution of individual titres. The GMT for BK-SE36/CpG arm was 83.2 (95% CI 47.7, 145.1) vs. GMT for control arm = 20.1 (95% CI 10.6, 38.1), *p* = 0.003 using the Wilcoxon test. 

Fluctuations in antibody titres following each vaccination and at post trial follow-up are shown in Figure 3B. A robust antibody response in the BK-SE36/CpG arm was observed 28 days after vaccination. Geometric mean anti-SE36 antibody levels in the BK-SE36/CpG arm were higher than in the control arm at all timepoints post-vaccination. In the control arm, the anti-SE36 IgG antibody titres remained relatively unchanged throughout. The highest antibody titres were observed after Dose 3. By Day 365, antibody titres had fallen somewhat but remained substantially higher than at D0 (before first vaccination) (Figure 3C). At Y3, titres remained higher than at D0 (*p* < 0.001); but lower than at D140 (*p* < 0.001) and D365 (1 year after Dose 1) (*p* = 0.0015). 

### 3.3. Factors Affecting Post Immunisation Immune Responses

#### 3.3.1. Baseline Anti-SE36 IgG Antibodies in the BK-SE36/CpG Arm

To address the question of whether sero-positivity (i.e., the presence of pre-existing antibodies) influences the response to vaccination, the BK-SE36/CpG arm was divided into two groups: one with detectable anti-SE36 IgG titres (vaccinees with titres > 8) at D0 and one without (vaccinees with titres below the limit of detection and, therefore, assigned a value of 8). Children in the “Value = 8 group” at D0 had substantially larger fold changes at D140 and D365 (Table 3) and a slightly larger fold change at Y3 compared to “Value > 8 group.” Children with pre-existing anti-SE36 IgG titres at D0, however, still tended to have higher geometric mean titres at D140 (post Dose 3), D365 (1 year after Dose 1) and Y3 (3 years after Dose 1) compared to those in the V = 8 group. The differences were not significant. 

#### 3.3.2. Participants’ Sex

The GMTs at the different timepoints were compared between male and female vaccinated children (Table 4). In terms of geometric mean titres, fold change from Day 0, and the proportion of subjects with detectable antibody titres, there were no marked differences in male and female children vaccinated with BK-SE36/CpG. In the present cohort, sex did not appear to influence the immune response.

## 4. Discussion

In the recently reported phase 1b trial, exploratory analysis of time-to-first clinical malaria episode in children 5 to 10 years old showed promising results from 28 days post Dose 2 to Day 365. Clinical efficacy was only an exploratory objective in the primary trial. In several sero-epidemiological studies in semi-immune adults or children, high anti-SE36 antibody titres correlated with the absence of fever, low parasitaemia, reduced clinical symptoms associated with severe malaria (coma, multiple convulsions and renal failure), reduced placental malaria, and normal newborn birthweight in pregnant women (reviewed in [30]). In a phase 1b trial in 6–20-year-old Ugandan participants, malaria incidences, measured as high parasitaemia (≥5000 parasites/µL whole blood) and fever (≥37.5 °C), were lower in BK-SE36 vaccinees compared to the control arm [31]. Although correlates of protection and analysis for cellular immune responses are lacking, studies on antibody level dynamics after vaccination and over time would also give important insights towards the efficacy and long-term protection of SE36-based vaccines. Here we show the persistence of anti-SE36 antibody titres following BK-SE36/CpG vaccination. 

Three years post Dose 1, the BK-SE36/CpG arm still had higher antibody titres compared to the rabies vaccine arm (control arm). In contrast to what appears to be a rapid decline in antibody titres after Dose 2 [38], there seems to be a slow decay from Day 365 to Y3. Between D140 to D365 (around 32 weeks) there was an 88% decrease in the peak titres observed post Dose 2. Between D365 to Y3 (around 32 months) there was a further 68% decrease in anti-SE36 antibody titres, reflecting a relatively “slower” decline. The proportion of children with detectable antibody titres decreased from 100% (D365) to 83% (Y3), in contrast to the control arm where detectable antibody titres at D365 and Y3 were at 47%. 

Notably, with another blood-stage vaccine candidate, MSP3-GLURP (GMZ2), the results are mixed with regards to the longevity of the humoral response. In malaria-naïve adults, anti-GLURP and anti-MSP3 antibodies were at significantly high levels after one year, although considerably lower compared to titres after Dose 2 and Dose 3 [41]. However, in phase 1b trials in healthy adults [42] and in children from 1 to 5 years of age [43] from Lambaréné, Gabon, after one year there was no significant difference in anti-GMZ2 IgG titres compared to baseline or between the vaccinated and control arms. In the proceeding phase 2 clinical trials in 1–5-year-olds in Burkina Faso, anti-GMZ2 antibodies in vaccinated children increased up to eight-fold from baseline (a 14-fold increase for 1–2 years old and a 5.7-fold increase in 3–4 years old) after Dose 3 [44]. There was no evidence of a decline in antibody titres in the 6-month follow-up period [44] but authors inferred a waning of antibody titres in a 2-year follow-up study because there was no evidence of protection (but no immunogenicity assessment was performed at this timepoint) [22]. 

Exposure to *P. falciparum* during the seasonal malaria transmission season may have contributed to sustaining the antibody titre in vaccinees. The observed sero-positivity at Day 0 in some children is supported by epidemiological studies showing some modest acquisition of anti-SE36 antibodies as a result of natural infection (reviewed in [30]). We cannot assess the role of natural infection in boosting antibody titres as reported in the Ugandan study [31] since no monitoring for malaria infection or frequent sampling for assessment of antibody titres were undertaken after the end of the primary trial (D365). However, considering that in the study area, malaria incidence was never below 13.7 cases/10,000 person-weeks and that the study participants were residing within a high malaria risk area in Ouagadougou [45], children are presumed to have had some exposure to natural infection during the three high malaria transmission seasons from December 2019 to March 2022. Thus, boosting by natural infection is likely to have occurred. The first-in-human BK-SE36/CpG trial in malaria-naïve Japanese adults (in the absence of natural malaria transmission) showed that the induced antibody titres remained above baseline during the 1-year follow-up but there was a 95% decline in GMTs compared to titres post Dose 2 [34].

Large variations were also observed in individual antibody titres. Independent of the immune mechanisms that are activated by vaccination, a number of associated factors could also affect the production of antibodies. These can be intrinsic (e.g., sex) and extrinsic (e.g., pre-vaccination titres or infection) [22,30,46,47,48,49,50]. In a previous BK-SE36 trial (formulation with only aluminium hydroxide gel as adjuvant) in Banfora, Burkina Faso, concurrent *P. falciparum* infection during vaccination in 12–60-month-old children resulted in lower humoral response compared to the immune response in uninfected children [51]. Interestingly, in the BK-SE36/CpG arm, the GMT measured in children with (sero-positive) or without (sero-negative) pre-existing titres were similar at D140, D365 and Y3. High anti-SE36 antibody titres were observed in vaccinees. However, the fold change from baseline was markedly different. Although a fold change of >2-fold from baseline was noted in all BK-SE36/CpG vaccinees post Dose 2 (D56) and Dose 3 (D140), those with pre-existing titres had lower fold change compared to children with undetectable antibody titres (assigned a value of 8) at baseline (D0). The association of pre-existing antibody titres with lowered immunogenicity in adults and children was also reported in other candidate malaria vaccine trials [30]. 

The use of CpG-ODN (K3) adjuvant appeared to overcome or break the immune tolerance in vaccinees with pre-existing antibody titres. In the Ugandan study, no increase in antibody titres or seroconversion (anti-SE36 IgG titres increased by >2-fold from baseline) were observed after two doses of BK-SE36 in sero-positive compared to sero-negative adults (reviewed in [30]). The pre-existing antibody titres in Ugandan adults were presumed to have overshadowed the immunogenicity of the vaccine. 

The persistence of the immunological response may also be attributed to the use of CpG-ODN (K3). At D365, no child had titres that dropped below the limit of detection. At Y3, 80% of the children (n = 12/15) with no detectable antibody titres and 64% of the children (n = 9/14) with detectable antibody titres at D0 had >2-fold change from baseline vs. 27% (n = 4/15) of the children in the control arm that sero-converted (i.e., having ≥2-fold increase in antibody titre from baseline). TLRs [52], specifically CpG-ODN (K3) based adjuvants, were inferred in other clinical trials to also lead to the development of long-lasting immunological memory [53,54]. 

An important factor to be defined in larger trials would be the threshold level of antibody titres able to provide protection against clinical malaria. Evaluation of the association between persistence of anti-SE36 antibodies and duration of protection will require further investigations. As shown in phase 3 trials with RTS,S/AS01, an association between the vaccine-induced anti-CSP antibodies and protection was observed. Although further studies suggested the importance of multiple mechanisms, the levels of anti-CSP were predictive of the duration of protection [55,56]. In RTS,S trials, the decay of anti-CSP antibody titres had a biphasic exponential distribution with a rapid waning in the first 6 months followed by a slower waning over the following 4 years [55]. Interestingly, waning antibody titres tend to predict the duration of efficacy: efficacy against clinical malaria declines more quickly in high-transmission regions due to decreased blood-stage immunity as well as decreased antibody titres. In low-transmission areas, vaccine efficacy wanes due to decreased anti-CSP antibody titres [55].

This single-visit cohort study has several limitations (e.g., small sample size, generally low infection rates, absence of infection data between Year 1 and Y3, absence of in vitro functional antibody results to correlate with antibody titre). Additionally, no follow-up was performed in the youngest cohort (12–24-month-old children) that participated in the primary trial. These limitations can be addressed in future larger clinical trials with improved formulation of SE36. However, at the same time, this study provides a longitudinal picture of the antibody level dynamics after vaccination and over time (>24 months after vaccination, in the absence of booster vaccination) in a malaria-endemic area. The findings provide support for future trials to demonstrate vaccine efficacy, the boosting of antibody titres as a result of natural infection, and dose optimization. Follow-up assessments for more than a year should be considered.

## 5. Conclusions

The malaria vaccine candidate BK-SE36/CpG induced high levels of antibodies that persisted up to Y3. Within the BK-SE36/CpG arm, the GMT at D140, D365, and Y3 of children with or without pre-existing antibody titres were largely similar at each timepoint. Sex-based differences in antibody production were not significant, although a tendency for higher GMT was observed in male children. Anti-SE36 antibody titres in BK-SE36/CpG vaccinees did decline considerably over time but remained above baseline (prior to Dose 1) and higher than in children in the control arm.

## Figures and Tables

**Figure 1 vaccines-12-00166-f001:**
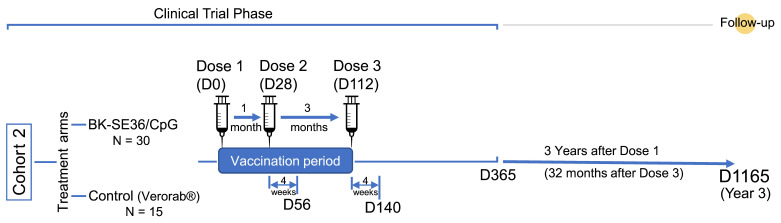
Scheme of the study. Number of subjects (N) and relevant timelines (D, days).

**Figure 2 vaccines-12-00166-f002:**
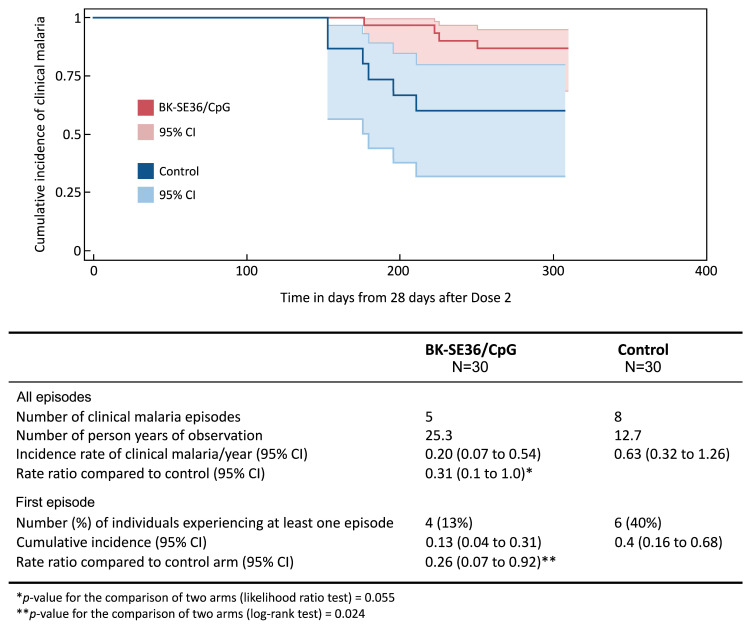
Incidence of clinical malaria. The Kaplan–Meier plot shows the time to the first episode of clinical malaria, defined as asexual parasitaemia ≥5000/µL whole blood and fever (≥38 °C), from 28 days post Dose 2 (Day 56) until the final visit (Day 365) between BK-SE36/CpG and control using a log rank test. The 95% confidence intervals (CI) for the risk of at least one episode were computed using Greenwood’s formula. In addition, the incidence rates of all episodes were calculated for each trial arm as the total number of episodes/number of person–years observed. The 95% confidence intervals were computed using random effects Poisson regression to account for possible repeated episodes within the same participant. Rate ratios were estimated using Cox regression with frailty to account for repeated episodes within the same participant.

**Figure 3 vaccines-12-00166-f003:**
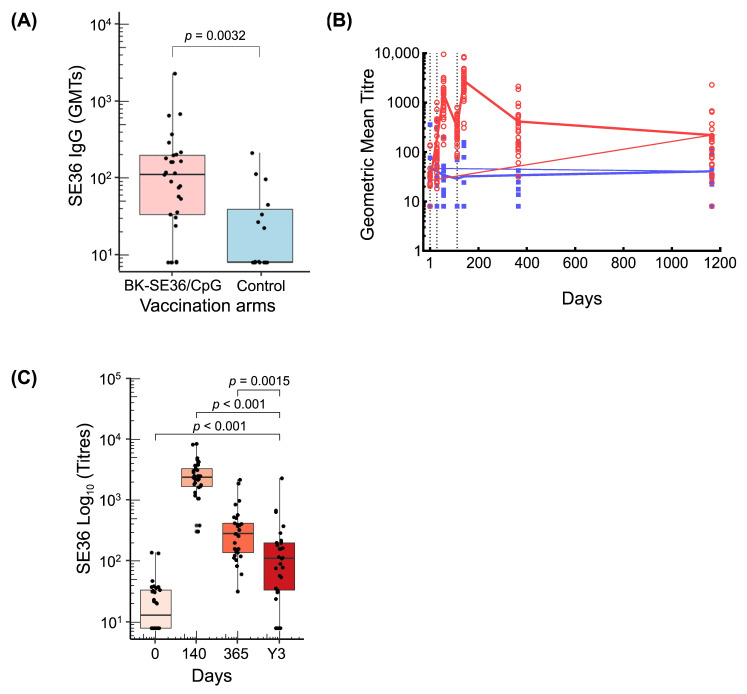
Anti-SE36 antibody responses. (**A**) Boxplot shows the distribution of individual titres, median titre, and interquartile range at Y3 in BK-SE36/CpG and control arm. (**B**) Line graph of GMT during the trial and follow-up period. Red circles and blue squares represent the values for individual subjects. Black dotted vertical lines correspond to vaccination days (Day 0, Day 28, Day 112). The last follow-up visit of the trial was at Day 365 and an additional single follow-up visit was conducted at Y3 (refer to Figure 1). (**C**) Boxplots of antibody titres from BK-SE36/CpG vaccinees at different timepoints (D140; D365 and Y3). BK-SE36/CpG: red; control: blue.

**Table 1 vaccines-12-00166-t001:** Characteristics of study population and malaria control measures practiced at Y3.

	D0 *		Y3	
	BK-SE36/CpG	Control	BK-SE36/CpG	Control
	N = 30	N = 15	N = 29	N = 15
Mean age (SD)	7.4 (1.1)	8.1 (1.3)	10.6 (1.1)	11.3 (1.3)
Sex (male/female, % male)	15/30 (50)	8/15 (53)	15/29 (52)	8/15 (53)
Proportion of participants with malaria positive slides, n/N (%)	1/30 (3)	1/15 (7)	0/29	0/15
Reported use of treated bed net, n/N (%)				
Last night			15/29 (52)	9/15 (60)
Last 3 months			20/29 (69)	10/15 (67)
Reported use of aerosol/repellent, n/N (%)				
Last night			1/29 (3)	0/15
Last 3 months			4/29 (14)	3/15 (20)
Reported use of mosquito coil, n/N (%)				
Last night			11/29 (38)	3/15 (20)
Last 3 months			14/29 (48)	7/15 (47)
Reported burning of herbs for malaria control, n/N (%)				
Last night			1/29 (3)	0/15
Last 3 months			1/29 (3)	0/15
Reported use of prophylactic medicine, n/N (%)				
Last night			0/29	0/15
Last 3 months			0/29	0/15

* No data were collected for malaria control measures practiced by participants at Day 0 [38].

**Table 2 vaccines-12-00166-t002:** Proportion of children with detectable anti-SE36 total IgG antibody titres.

Timepoints	BK-SE36/CpG n/N (%)	Control n/N (%)	*p*-Value *
D0 (before Dose 1)	15 (50)	9 (60)	
D365 (1 year after Dose 1)	30 (100)	7 (47)	<0.001
Y3 (3 years after Dose 1)	24 (83)	7 (47)	0.039

* Using the Wilcoxon rank-sum test.

**Table 3 vaccines-12-00166-t003:** Anti-SE36 IgG titres and fold change in BK-SE36/CpG vaccinees with or without pre-existing antibody titres prior to Dose 1.

	SE36 Antibody Titres at Day 0 (prior to Dose 1)			
	Value = 8 *	Value > 8	Total	
Timepoints	N = 15	N = 15	N = 30	*p*-Value **
Day 140 vs. Day 0				
Geometric mean titre (95% CI), D140	2070 (1268, 3378)	2405 (1789, 3233)	2231 (1704, 2922)	0.9674
Geometric mean fold change in titre (95% CI)	258.7 (158.5, 422.3)	64.5 (43.4, 95.8)	129.2 (86.9, 191.9)	0.0002
Day 365 vs. Day 0				
Geometric mean titre (95% CI), D365	231.1 (143.9, 370.9)	300.7 (168.2, 537.6)	263.6 (184.9, 375.8)	0.6827
Geometric mean fold change in titre (95% CI)	28.9 (18.0, 46.4)	8.1 (4.8, 13.5)	15.3 (10.2, 22.9)	0.0007
	**N = 15**	**N = 14**	**N = 29**	
Year 3 vs. Day 0				
Geometric mean titre (95% CI), Y3	59.5 (30.4, 116.4)	119.2 (45.8, 310.7)	83.2 (47.7, 145.1)	0.2690
Geometric mean fold change in titre (95% CI)	7.4 (3.8, 14.6)	3.2 (1.3, 8.2)	5.0 (2.8, 8.6)	0.1223

* A value of 8 was assigned to undetectable titres. ** Using the Mann–Whitney test.

**Table 4 vaccines-12-00166-t004:** Anti-SE36 IgG titres and proportion of male and female BK-SE36/CpG vaccinated children with detectable anti-SE36 antibodies at different timepoints.

	Sex			
	Male (95% CI)	Female (95% CI)	Total	
Timepoints	N = 15	N = 15	N = 30	*p*-Value *
D0 (before Dose 1)				
Geometric mean titre (95% CI)	16.1 (9.5, 27.1)	18.6 (11.7, 29.6)	17.3 (12.4, 24)	0.7261
Proportion of subjects with detectable titres	40%	60%	50%	
D140 (28 days after Dose 3)				
Geometric mean titre (95% CI)	2293 (1342, 3918)	2171 (1762, 2676)	2231 (1704, 2922)	0.5393
Geometric mean fold change from D0 (95% CI)	142.7 (73.6, 276.5)	116.9 (69.6, 196.6)	129.2 (86.9, 191.9)	0.6827
Proportion of subjects with detectable titres	100%	100%	100%	
D365 (1 year after Dose 1)				
Geometric mean titre (95% CI)	320.1 (187.4, 546.8)	217.1 (130.2, 361.9)	263.6 (184.9, 375.8)	0.1607
Geometric mean fold change from D0 (95% CI)	19.9 (10.0, 39.8)	11.7 (7.2, 18.9)	15.3 (10.2, 22.9)	0.1736
Proportion of subjects with detectable titres	100%	100%	100%	
	**N = 15**	**N = 14**	**N = 29**	
Y3 (3 years after Dose 1)				
Geometric mean titre (95% CI)	94.7 (41.6, 215.7)	72.4 (30.9, 170.1)	83.2 (47.7, 145.1)	0.8801
Geometric mean fold change from D0 (95% CI)	5.9 (2.5, 14.1)	4.1 (1.9, 9.1)	5.0 (2.8, 8.6)	0.5611
Proportion of subjects with detectable titres	87%	79%	83%	

A value of 8 was assigned to undetectable titres. * Using the Mann–Whitney test.

## Data Availability

The data presented in this study are available on request from the corresponding authors.

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
