# Peer review of "Persistence of Anti-SE36 Antibodies Induced by the Malaria Vaccine Candidate BK-SE36/CpG in 5–10-Year-Old Burkinabe Children Naturally Exposed to Malaria"

_vaccines, 2024, doi:10.3390/vaccines12020166_

Round 1

Reviewer 1 Report

Comments and Suggestions for Authors

The authors showed about the persistence of anti-SE36 antibodies in children living in Burkina Faso received with SE36/CpG even after 3 years of immunization.

minor comments about boost effect by naturally exposed malaria. Fluctuation of antibody against another malaria antigen, i.e. anti-MSP-1 should be examined and compared with anti-SE36 antibody.

Author Response

Reviewer 1:

The authors showed about the persistence of anti-SE36 antibodies in children living in Burkina Faso received with SE36/CpG even after 3 years of immunization.

minor comments about boost effect by naturally exposed malaria. Fluctuation of antibody against another malaria antigen, i.e. anti-MSP-1 should be examined and compared with anti-SE36 antibody.

R: We have used other antigens for comparison in our sero-epidemiological studies and consistently other antigens like MSP-1 show a different immune response. However, natural infections can only induce modest or no antibody against SE36 (Horii et al., 2010; Owalla et al., 2013: reviewed in Palacpac et al., 2024).  In this study we wanted to see the persistence of the anti-SE36 antibody titres after vaccination. Without having children vaccinated with another vaccine candidate, we could not really conclude strongly with regards to the fluctuation in antibody response or boosting effect as a result of natural infection using another antigen.  Notably, while it is indeed ideal to use another antigen for comparison, unfortunately we do not have any other blood-stage candidate vaccine at our disposal for the trial. In the manuscript we have made relative comparisons with the results that were reported for the GMZ2 trial.  Sentences were inserted in Lines 285-297.

Reviewer 2 Report

Comments and Suggestions for Authors

Please find below review comments for the manuscript entitled “Persistence of anti-SE36 antibodies induced by the malaria vaccine candidate BK-SE36/CpG in 5-10 year old Burkinabe children naturally exposed to malaria”

Line 17: A recent vaccine trial in healthy malaria exposed African adults and children…

Please consider if ‘previously malaria exposed but healthy African adults and children’ will be more appropriate here.

Line 37: This has highlighted the plasticity….

A better wording would be …’This has highlighted the difficulties associated with malaria elimination as well as the challenges….

Line 86: Please define (K3) in CpG-ODN (K3).

Line 104: Study design and participants.

It will be a good idea to include a simple timeline by drawing an arrow (Figure), showing all details of number of subjects, immunization time points and follow-up time points from day 0 to 3 years  (Lines 109-113; 178-186, etc). This will help capture all the important time points and events of the study. Also, it will help to keep the scales of figures consistent, either in days or weeks.

Line 118: Each child enrolled should have satisfied the following inclusion criteria.

The statement should be in the past tense – ‘Each enrolled child met the following inclusion criteria’, or…’The inclusion criteria included the following conditions’

Line 121: None should have fulfilled any of the exclusion criteria.

Please rephrase as…’The exclusion criteria included the following conditions’

Line 128: Please mention full form of GRAS

Line 160: Please provide full form of SMC

Line 166: Clinical malaria (defined as ≥ 5000 parasites/μL + fever, ≥ 38°C),

Please define as ≥ 5000 parasites/μL of whole blood, or packed red blood cells etc…

Figure 1: It will be better to keep the X-axis scale in days or weeks. The X-axis needs to be extended at least to the pouts where the data is available and shown in the figure (Currently it is 0.8 years on the X-axis but the data extends beyond that.  

Line 190: Some participants (20%) did not practice any malaria control measures.

Change to ‘About 20% participants did not practice any malaria control measures’.

Figure 2:

Please show baseline or cut-off value line in graph where applicable.

Line 206: The last follow-up visit of the trial was at Day 365 and a single follow-up visit was conducted at Y3.

Better to rephrase as ‘The last follow-up visit of the trial was at Day 365 and a an additional single follow-up visit was conducted at year 3.

Line 207-208:

Please keep the time points description consistent – either in days or weeks (but not mixed).

Table 3:

Under ‘Time points’, the bold headings such as ‘Fold change D140/D0’ appears to need correction since there are both the ‘titers’ and ‘fold change’ below it. Please change ‘Fold change’ to ‘Day 140 vs Day 0’ etc.

Line 240: Please remove ‘importantly’.

Lines 281-283: Independent of the immune mechanisms that is activated by vaccination, it is known that a number of associated factors could affect the production of antibodies.

Please rephrase as ‘Independent of the immune mechanisms that are activated by vaccination, a number of associated factors could also affect the production of antibodies’.

A description in brief of the BK-SE36/CpG formulation and dose amount would be helpful.

While this study provided an insight into the persistence of antibody levels against the vaccine candidate BK-SE36/CpG in 5-10 year olds, any correlation of this antibody level in vaccinees with infection at year 3 time point compared to control group would add value to the overall data.

Comments on the Quality of English Language

Quality of English is fine, some minor edits may be required.

Author Response

Reviewer 2:                                                                                                         

Please find below review comments for the manuscript entitled “Persistence of anti-SE36 antibodies induced by the malaria vaccine candidate BK-SE36/CpG in 5-10 year old Burkinabe children naturally exposed to malaria”

Line 17: A recent vaccine trial in healthy malaria exposed African adults and children…

Please consider if ‘previously malaria exposed but healthy African adults and children’ will be more appropriate here.

R: Line 17 revised to “malaria-exposed, healthy African adults and children”.

Line 37: This has highlighted the plasticity….

A better wording would be …’This has highlighted the difficulties associated with malaria elimination as well as the challenges….

R: Revised the sentence. The difficulties associated with malaria elimination is not only restricted to the lack of tools/interventions but is also attributable to the phenotypic plasticity of both the mosquito and the parasite.  The revised sentence now reads: The difficulties in malaria elimination have highlighted the plasticity of the mosquito and Plasmodium parasite, as well as the challenges in the implementation of these control strategies in the field.  Revised Lines 37-40.

Line 86: Please define (K3) in CpG-ODN (K3).

R: Revised in Lines 88-92, with additional information from previous studies: K3 (ATCG ACTC TCGA GCGT TCTC) is a novel K-type synthetic oligodeoxynucleotide with unmethylated CpG motif (Verthelyi et al., 2001, 2002, 2003). When used concomitantly with aluminum hydroxide gel, K3 was chosen as the best Toll-like receptor 9 agonist with the highest ability to induce antibody and cellular immune responses compared to BK-SE36 alone [Tougan et al., 2013].

Line 104: Study design and participants.

It will be a good idea to include a simple timeline by drawing an arrow (Figure), showing all details of number of subjects, immunization time points and follow-up time points from day 0 to 3 years  (Lines 109-113; 178-186, etc). This will help capture all the important time points and events of the study. Also, it will help to keep the scales of figures consistent, either in days or weeks.

R: Thank you for this suggestion. A new Figure 1 is created.  To be consistent, Fig. 2 was also revised; timelines were changed to days. 

Line 118: Each child enrolled should have satisfied the following inclusion criteria.

The statement should be in the past tense – ‘Each enrolled child met the following inclusion criteria’, or…’The inclusion criteria included the following conditions’

R: Revised as suggested. Lines 131-134

Line 121: None should have fulfilled any of the exclusion criteria.

Please rephrase as…’The exclusion criteria included the following conditions’

R: Revised as suggested. Lines 134-135

Line 128: Please mention full form of GRAS

R: Revised as suggested, Line 140

Line 160: Please provide full form of SMC

R: Revised as suggested, Line 174. In the previous ms, SMC was defined in Line 37. Retained the full form and abbreviation in Line 36 to be easily recognized by readers for malaria control interventions.  

Line 166: Clinical malaria (defined as ≥ 5000 parasites/μL + fever, ≥ 38°C),

Please define as ≥ 5000 parasites/μL of whole blood, or packed red blood cells etc…

R: Revised as suggested: “of whole blood.” Lines 180, 185, 270

Figure 1: It will be better to keep the X-axis scale in days or weeks. The X-axis needs to be extended at least to the pouts where the data is available and shown in the figure (Currently it is 0.8 years on the X-axis but the data extends beyond that. 

R: In the current ms version this is now Figure 2.  X-axis scale was changed to days and the axis extended to the date where the data is available.

Line 190: Some participants (20%) did not practice any malaria control measures.

Change to ‘About 20% participants did not practice any malaria control measures’.

R: Revised as suggested, Line 209

Figure 2:                                                                                                                              

Please show baseline or cut-off value line in graph where applicable.

R: Now Figure 3. Titres at Year 3 is shown in Fig 3A, comparing for both BK-SE36/CpG and control arm.  Fig 3B shows the fluctuation of antibody titers during the whole study, showing the peaks after each vaccination in both treatment arms (BK-SE36/CpG, red; control, blue). Fig 3C presents only the fluctuation in antibody titre for BK-SE36/CpG: at Day 0 (baseline), 4 weeks after Dose 3 (D140), at the end of the study (D365), and at Year 3.  Please reconsider.  We think that Fig 3A and 3B can show clearly the baseline or cut-off value and while we agree that it is not so clear in Fig 3B as the baseline and undetectable points are overlaid on each other-- that is also due to large variations in antibody titers in both the control and BK-SE36/CpG arms. Fig 3B nevertheless shows the antibody titer trends/changes from baseline to Y3, accommodating all datapoints (and thus we would like to keep this figure). Splitting Fig. 3B in both or either the x- and y-axis to accommodate all datapoints fail to show the trends and makes the graph complicated.

Line 206: The last follow-up visit of the trial was at Day 365 and a single follow-up visit was conducted at Y3.

Better to rephrase as ‘The last follow-up visit of the trial was at Day 365 and a an additional single follow-up visit was conducted at year 3.

R: Revised as suggested, Line 225

Line 207-208:

Please keep the time points description consistent – either in days or weeks (but not mixed).

R: Revised as suggested. To be consistent, timelines were chosen to be expressed as Days.

Table 3:

Under ‘Time points’, the bold headings such as ‘Fold change D140/D0’ appears to need correction since there are both the ‘titers’ and ‘fold change’ below it. Please change ‘Fold change’ to ‘Day 140 vs Day 0’ etc.

R: Revised as suggested, please see Table 3.

Line 240: Please remove ‘importantly’.

R: Revised as suggested, Line 257

Lines 281-283: Independent of the immune mechanisms that is activated by vaccination, it is known that a number of associated factors could affect the production of antibodies.

Please rephrase as ‘Independent of the immune mechanisms that are activated by vaccination, a number of associated factors could also affect the production of antibodies’.

R: Revised as suggested., Lines 314-316.

A description in brief of the BK-SE36/CpG formulation and dose amount would be helpful.

R: Revised as suggested in Lines 112-115.

While this study provided an insight into the persistence of antibody levels against the vaccine candidate BK-SE36/CpG in 5-10 year olds, any correlation of this antibody level in vaccinees with infection at year 3 time point compared to control group would add value to the overall data.

R: As mentioned in Line 198, at year 3 timepoint no parasite infection was found. The limitation with regards to the absence of infection data from Day 365 to Year 3 was also mentioned in Line 303.

Reviewer 3 Report

Comments and Suggestions for Authors

In this paper, Issa Nebie et al. followed the participant for a Phase I clinicla trial of BK-SE36/CpG  vaccine. They determined the change of IgG titer after vaccine and the clinical malaria event between the vaccine and control groups.  Even the limitation of sample size as the Phase I clinical trial, there is significantly less clinical disease in the vaccine group.  This study offers an important information about current developing RBC stage vaccine which should be known for the field.

Minor comments:

Table 2 should include the statistic method that bring the significance as the other tables. 

Line 139, the author should explain why they assign 8 as undetectable levels. 

In the control group, there are couple of participants had significant IgG titers at Day0. It will be valuable that the authors can offer the potential explaination.  

Author Response

Reviewer 3

In this paper, Issa Nebie et al. followed the participant for a Phase I clinicla trial of BK-SE36/CpG  vaccine. They determined the change of IgG titer after vaccine and the clinical malaria event between the vaccine and control groups.  Even the limitation of sample size as the Phase I clinical trial, there is significantly less clinical disease in the vaccine group.  This study offers an important information about current developing RBC stage vaccine which should be known for the field.

Minor comments:

Table 2 should include the statistic method that bring the significance as the other tables.

R: The analysis was done using Wilcoxon-Rank-Sum test. Added a footnote in Table 2.

Line 139, the author should explain why they assign 8 as undetectable levels.

R: Revised in Lines 150-153. The value of 8 is based on the mean (threshold) value of 60 samples (20 individual negative control sera from Japanese adults (malaria naïve) assayed in three independent batches).

In the control group, there are couple of participants had significant IgG titers at Day0. It will be valuable that the authors can offer the potential explaination. 

R: Inserted Lines 298-301.